# Experimental Polymorphism Survey in Intergenic Regions of the *icaADBCR* Locus in *Staphylococcus aureus* Isolates from Periprosthetic Joint Infections

**DOI:** 10.3390/microorganisms10030600

**Published:** 2022-03-10

**Authors:** Liliana Morales-Laverde, Maite Echeverz, Margarita Trobos, Cristina Solano, Iñigo Lasa

**Affiliations:** 1Laboratory of Microbial Pathogenesis, Navarrabiomed, Hospital Universitario de Navarra (HUN), Universidad Pública de Navarra (UPNA), IdiSNA, 31008 Pamplona, Spain; lilianaandrea.morales@unavarra.es (L.M.-L.); maite.echeverz@unavarra.es (M.E.); cristina.solano@unavarra.es (C.S.); 2Department of Biomaterials, Institute of Clinical Sciences, Sahlgrenska Academy at University of Gothenburg, 40530 Gothenburg, Sweden; margarita.trobos@biomaterials.gu.se

**Keywords:** periprosthetic joint infection, PIA/PNAG, *ica* operon, SNPs, IGRs, *Staphylococcus aureus*

## Abstract

*Staphylococcus aureus* is a leading cause of prosthetic joint infections (PJI) characterized by bacterial biofilm formation and recalcitrance to immune-mediated clearance and antibiotics. The molecular events behind PJI infection are yet to be unraveled. In this sense, identification of polymorphisms in bacterial genomes may help to establish associations between sequence variants and the ability of *S. aureus* to cause PJI. Here, we report an experimental nucleotide-level survey specifically aimed at the intergenic regions (IGRs) of the *icaADBCR* locus, which is responsible for the synthesis of the biofilm exopolysaccharide PIA/PNAG, in a collection of strains sampled from PJI and wounds. IGRs of the *icaADBCR* locus were highly conserved and no PJI-specific SNPs were found. Moreover, polymorphisms in these IGRs did not significantly affect transcription of the *icaADBC* operon under in vitro laboratory conditions. In contrast, an SNP within the *icaR* coding region, resulting in a V176E change in the transcriptional repressor IcaR, led to a significant increase in *icaADBC* operon transcription and PIA/PNAG production and a reduction in *S. aureus* virulence in a *Galleria mellonella* infection model. In conclusion, SNPs in *icaADBCR* IGRs of *S. aureus* isolates from PJI are not associated with *icaADBC* expression, PIA/PNAG production and adaptation to PJI.

## 1. Introduction

Despite the efforts at improving infection control measures to reduce the risk of bacterial colonization of arthroplasty implants, the incidence of periprosthetic joint infections (PJI) remains high [1,2]. *Staphylococcus aureus* is one of the leading pathogens causing PJI due to its high capacity to adhere to the surface of the implant and persist in a self-produced extracellular matrix [3,4]. The attachment of *S. aureus* to the surface of the medical device can occur at the time of surgery or after implantation, once the surface is covered with plasma proteins. In the former case, adhesion of the bacteria to the naked surface is dependent on the physico-chemical characteristics of the device and the bacterial surface components [5]. In the later, attachment to the implanted device mainly occurs through specific interactions between bacterial adhesins and plasma proteins [6,7]. Once attached, *S. aureus* forms a biofilm due to the production of an extracellular matrix mainly composed of exopolysaccharides, proteins, and extracellular DNA [8,9]. As a result, the infection is difficult to treat because bacteria inside the biofilm grow protected from the attack of the immune system and antimicrobial agents [10,11].

The major exopolysaccharide of the *S. aureus* biofilm matrix is a polymer of β-1,6-linked *N*-acetylglucosamine named PIA (polysaccharide intercellular adhesin) or PNAG (poly-*N*-acetyl-glucosamine) [12]. Production of the PIA/PNAG exopolysaccharide is dependent upon the proteins encoded by the *icaADBC* operon whose expression is regulated by the transcriptional repressor IcaR [13] and other regulatory elements outside the *ica* locus [14,15,16,17,18]. Several of these regulatory proteins (IcaR, TcaR, SarA, ArlR) have been shown to bind to the intergenic region (IGR) between *icaADBC* and *icaR* to modulate *icaADBC* expression [14,15,17,19]. The fact that so many different regulatory proteins bind to the same promoter region strongly suggests that PIA/PNAG expression has to be tightly controlled so that a biofilm is formed in a timely and appropriate manner according to changing environments.

It has been a long-held desire to identify biofilm phenotypic characteristics or molecular markers that predict the propensity of a *Staphylococcus* spp. isolate to produce PJI. However, these efforts have been unsuccessful [20,21]. Several reasons could explain the failure in finding adaptative signatures: (i) the in vitro biofilm assays may not resemble the biofilms’ characteristics found in PJI; (ii) the capacity to produce PJI might depend on the levels of many regulatory proteins that directly or indirectly modulate the expression of adhesins and biofilm matrix compounds [22]; (iii) differences in the capacity to produce PJI might be based on nucleotide polymorphisms (SNPs) in the regulatory IGRs of the genes encoding for biofilm matrix compounds instead of inside coding sequences; and, (iv) the *S. aureus* strains producing PJI may not have any specific adaptation to produce PJI.

As regards SNPs in IGRs, their high frequency together with the difficulties in anticipating their relevance without experimental analysis have biased most of the studies of bacterial adaptation to infection towards coding regions [23]. However, it has been shown that genetic variations in IGRs can have important consequences for the adaptation of *Salmonella typhimurium*, *Pseudomonas aeruginosa* and *Mycobacterium tuberculosis* to the environmental conditions of the host [24,25,26,27,28]. In *S. aureus*, intergenic SNPs have been found to be significantly associated with toxicity in methicillin-resistant strains (MRSA) [29] and highly divergent IGRs have been related to differences in gene expression [30]. Moreover, a naturally occurring 5 nt deletion in the promoter region of the *icaADBC* operon in *S. aureus* MN8 has been shown to cause a drastic increase in PIA/PNAG levels [31].

In this study, we aimed to identify SNPs in the regulatory IGRs of the *icaADBCR* locus in the genomes of a collection of *S. aureus* isolates from PJI, and to analyze their potential association with the propensity to cause arthroplasty infection. Using reporter fusions of representative IGRs with the green fluorescent protein encoding gene, we explored the contribution of SNPs in IGRs to *icaADBC* operon transcription. Also, PIA/PNAG production and biofilm formation capacity of representative isolates was examined. Our results show high sequence conservation of IGRs, suggesting that the necessity to tightly regulate PIA/PNAG expression imposes constraints to IGRs of the *icaADBCR* locus. We further provide evidence that the SNPs found in PJI isolates are not associated with changes in *icaADBC* expression and the capacity of the strains to synthesize PIA/PNAG, form a biofilm and cause PJI.

## 2. Materials and Methods

### 2.1. Bacterial Strains, Plasmids, Oligonucleotides and Culture Conditions

Bacterial strains, plasmids, and oligonucleotides used in this work are listed in Table 1 and Table 2. The forty-three PJI *S. aureus* strains used in this study were isolated from patients admitted at Sahlgrenska University Hospital (Mölndal, Sweden) with PJI of the hip or knee [32,33]. On the other hand, twenty-six strains from wound swabs, isolated between 1966 and 2010, were obtained from the Culture Collection University of Gothenburg (CCGU).

*Escherichia coli* and *S. aureus* strains were routinely grown in Luria-Bertani medium (LB; Conda-Pronadisa, Madrid, Spain) and Trypticase soy broth (TSB; Conda-Pronadisa, Madrid, Spain), respectively, at 37 °C. When required, media were supplemented with appropriate antibiotics at the following concentrations: erythromycin (Ery), 10 µg/mL or 2.5 µg/mL; ampicillin (Amp), 100 µg/mL. Bacteriological agar was used as a gelling agent (VWR, Radnor, PA, USA).

### 2.2. DNA Manipulations

Routine DNA manipulations were performed using standard procedures unless otherwise indicated [39]. Oligonucleotides were synthesized by STABVIDA, Lda. (Caparica, Portugal). FastDigest restriction enzymes, Phusion DNA polymerase, and the rapid DNA ligation kit (Thermo Scientific, Waltham, MA, USA) were used according to the manufacturer’s instructions. Plasmids were purified using a plasmid purification kit from Macherey-Nagel (Allentown, PA, USA) according to the manufacturer’s protocol. Plasmids were transformed in *E. coli* by electroporation (1 mm cuvette; 200 Ω, 25 µF, 1250 V; Gene Pulser X-Cell electroporator). *S. aureus* competent cells were generated as previously described [40]. Plasmids were transformed in *S. aureus* by electroporation (1mm cuvette; 100 Ω, 25 µF, 1250 V; Gene Pulser X-Cell electroporator). All constructed plasmids were confirmed by Sanger sequencing at STABVIDA Lda. (Caparica, Portugal).

To construct the MW2 *icaR* V176E mutant, two DNA fragments were amplified with the primer pairs Au56/LM9 and LM10/Au73 (Table 2) from the MW2 wild type strain. The two PCR fragments were fused through overlapping PCR using primers Au56 and Au73, cloned into the pJET 1.2 vector and then subcloned into the pMAD vector [37] digested with SalI and BamHI, generating plasmid pMAD::*icaR*V176E. This plasmid was purified from *E. coli* IM01B and transformed into the MW2 wild type strain by electroporation. Homologous recombination experiments were performed as described [14]. Erythromycin-susceptible white colonies, which did not further contain the pMAD plasmid, were tested by PCR using the primers Au56 and Au73 and further digestion of the PCR product with ScaI.

### 2.3. Whole-Genome Sequencing and Genomic Analysis

Whole genome sequencing of PJI isolates was performed previously [32,33]. Genomic DNA from the twenty-six *S. aureus* strains isolated from wounds was extracted with the GenEluteTM Bacterial Genomic DNA kit (Sigma-Aldrich, St Louis, MO, USA) according to the manufacturer’s instructions for Gram-positive bacteria. DNA was used for library preparation using a Nextera XT DNA Library Prep Kit (Illumina, San Diego, CA, USA) followed by Illumina sequencing at MicrobesNG (University of Birmingham, UK) on a HiSeq2500 Illumina sequencer using a paired-end approach (2 × 250 bp). The Sequence Read Archives (SRAs) with detailed information of the strains was deposited in the National Center for Biotechnology Information (NCBI) under the BioProject accession number PRJNA765573.

Nodes containing the *ica* locus were localized in each strain using the blastn tool from the NCBI website. Analysis of the SNPs and comparison between the strains were performed with SnapGene v5.3.3. All sequences containing the *ica* locus were aligned and compared against the *S. aureus* MW2 reference genome (GenBank accession number NC_003923). Genetic analysis was focused on the regulatory sequences of the *icaADBC* operon including the 3′UTR of *icaR*, the *icaR* coding sequence, and the IGR between *icaR* and *icaA*.

### 2.4. Generation of Transcriptional Fusions of Ica Regulatory Sequences with gfp

To generate transcriptional fusions of *ica* regulatory sequences with the *gfpmut2* gene, the region comprising the 3′UTR of *icaR*, the *icaR* coding sequence, and the IGR between *icaR* and *icaA* was amplified from each representative strain of clusters one to six using the primers Au56 and Au76 (Table 2). The PCR products were cloned into the pJET 1.2 vector and then subcloned into the pCN52 plasmid [38] digested with SalI and KpnI, giving plasmids pCN52::IGR*ica* C1 to C6. To generate transcriptional fusions of *ica* regulatory sequences containing a mutated *icaR* gene, with a premature stop codon after residue 36, two DNA fragments were amplified with primer pairs Au56/Au78 and Au77/Au76 (Table 2) from each representative strain of clusters one to six. The two PCR fragments were fused through overlapping PCR using primers Au56 and Au76, cloned into the pJET 1.2 vector and then subcloned into pCN52 digested with SalI and KpnI, giving plasmids pCN52::IGRStop*ica* C1 to C6.

### 2.5. Western Blotting

Overnight cultures of *S. aureus* 132 *ica::tet* transformed with pCN52::IGR*ica* or pCN52::IGRStop*ica* plasmids were diluted 1:100 and grown in TSB supplemented with 3% NaCl (TSB-NaCl) at 37 °C under static conditions. Samples were collected when the OD 590_nm_ reached a value of 0.7. Cells were re-suspended in phosphate buffered saline (PBS) and lysed using a FastPrep-24™ 5G (MP Biomedicals, LLC, Irvine, CA, USA). Supernatants from total protein extracts were recovered and analysed by SDS-PAGE and Western blotting, as follows. A volume of Laemmli buffer was added to the samples and boiled for 5 min; 4 µg of protein was used for SDS-PAGE analysis with the TGX™ Stain-Free FastCast™ Acrylamide Kit, 12% (Bio-Rad, Hercules, CA, USA). For Western blot analysis, protein extracts were blotted onto Amersham^TM^ Protran^TM^ Premium 0.45 µm nitrocellulose blotting membranes (Cytiva, Marlborough, MA, USA) by electroblotting. Membranes were blocked overnight in PBS containing 0.1% Tween 20 and 5% skimmed milk under shaking conditions, and incubated with anti-GFP antibodies (Living Color A.v. monoclonal antibody JL-8 Clontech, Mountain View, CA, USA), diluted 1:2500 in blocking solution for 2 h at room temperature. Alkaline phosphatase conjugated goat anti-mouse immunoglobulin G (Sigma-Aldrich, St Louis, MO, USA), diluted in blocking solution, was used as a secondary antibody, and the subsequent chemiluminescence reaction was recorded with ECL Prime western blotting detection reagents (Cytiva, Marlborough, MA, USA) in a ChemiDoc MP Imaging System (Bio-Rad, Hercules, CA, USA).

### 2.6. Biofilm Assay and PIA/PNAG Detection

The biofilm formation assays were performed in sterile 96-well polystyrene microtiter plates (Thermo Scientific, Waltham, MA, USA) as described elsewhere [41]. Briefly, overnight cultures grown in TSB were diluted 1:100 in TSB-NaCl and incubated at 37 °C for 24 h. The wells were gently rinsed three times with water, dried, and stained with 0.1% crystal violet for 15 min. The wells were rinsed again, and the crystal violet was solubilized with 200 µL of ethanol-acetone (80:20, *vol*/*vol*). The OD 590_nm_ was determined in an Epoch (BioTek, Winooski, VT, USA) microplate spectrophotometer. Biofilm formation experiments were performed in triplicate (*n* = 3) with three technical replicates.

PIA/PNAG was extracted and detected by dot-blot as described elsewhere [13]. Overnight cultures were diluted 1:100 in 2 mL TSB-NaCl and incubated at 37 °C for 24 h in sterile 24-well cell culture plates from Costar (Corning, New York, NY, USA). Biofilm cells were recovered from each well, centrifuged and suspended in 50 µL of 0.5 M EDTA (pH 8.0). Cells were then incubated for 5 min at 100 °C and centrifuged. The supernatant (40 µL) was incubated with 10 µL of proteinase K (20 mg/mL) for 30 min at 37 °C. After the addition of 10 µL of Tris-buffered saline (20 mM Tris-HCl, 150 mM NaCl [pH 7.4]) containing 0.01% bromophenol blue, 4 µL were spotted onto Amersham^TM^ Protran^TM^ Premium 0.45 µm nitrocellulose blotting membranes (Cytiva, Marlborough, MA, USA) using a Bio-Dot microfiltration apparatus (Bio-Rad, Hercules, CA, USA), dried and blocked overnight. The membrane was incubated with an anti-PNAG antibody diluted 1:10,000 at room temperature for 2 h [42]. Bound antibodies were detected with peroxidase-conjugated goat anti-rabbit immunoglobulin G antibodies (Jackson ImmunoResearch Laboratories, Inc., Westgrove, PA, USA).

### 2.7. Galleria Mellonella Survival Assay

*Galleria mellonella* larvae were obtained in bulk from Bichosa (Vigo, Spain), and stored at 16 °C until used. Bacterial cultures were grown in TSB for 16 h at 37 °C under shaking, and cultures were pelleted and suspended in 2 mL of PBS to obtain a cell density of 1 × 10^9^ CFU/mL. For infection, 10 μL of the bacterial suspensions (1 × 10^7^ CFU/larvae) were injected in the left lower abdominal proleg with a Micro Fine Insulin Syringe 0.3 mL, 30g (BD, Frajklin Lakes, NJ, USA), using a Burkard hand microapplicator (Bukard Scientific, Rickmansworth, UK). The control group was injected with 10 µL of sterile PBS. A group of 25 larvae was used for each strain. Infected larvae were transferred to a clean Petri dish lined with filter paper, incubated at 37 °C, and scored for survival every 24 h for a total of 96 h. The survival experiments were performed with three biological replicates.

### 2.8. Statistical Analyses

Statistical analyses were performed with the GraphPad Prism v9.2.0 program. For single comparisons, data were analysed using an unpaired, two-tailed Student’s *t*-test. Data corresponding to the *Galleria mellonella* survival assay were compared using a log-rank (Mantel–Cox) test. In all tests, *p* values of less than 0.05 were considered statistically significant: * *p* < 0.05, ** *p* < 0.01, *** *p* < 0.001.

## 3. Results

### 3.1. The SNPs Occurrence Rate Is Low in the Regulatory IGRs of the icaADBCR Locus

Most studies of the host adaptation have focused on genetic variations within coding regions, whereas the role of SNPs in IGRs has remained disregarded [27]. We hypothesized that SNPs in the regulatory IGRs of the *icaADBCR* locus might reflect differences in the capacity of an *S. aureus* strain to produce implant-associated infections. To explore this hypothesis, we first compared the *ica* locus sequence conservation among 1000 complete genomes of *S. aureus* available at the NCBI database using blastn. The *ica* locus consists of the regulatory gene, *icaR*, located upstream and divergently transcribed from the biosynthetic operon *icaADBC*. For the analysis, the locus sequence was divided into the following segments: the 3′UTR of *icaR*, involved in the post-transcriptional regulation of *icaR* expression [43]; the coding sequence of *icaR*; the entire *icaR-icaADBC* IGR containing *icaR* and *icaADBC* promoters and 5´UTRs, and the coding sequences of *icaA*, *icaD*, *icaB*, and *icaC*. Also, the flanking IGRs outside of the *ica* locus were included (Figure 1). The results revealed a variation of 8.2% in the 3′UTR of *icaR*, 9.3% in the *icaR-icaA* IGR and variation rates that ranged from 3.4% to 12% in the *icaR* and *icaADBC* coding sequences. Contrary to this low degree of variation along the *ica* locus, the variation rates of the IGRs flanking the *icaADBCR* locus were 21.2% and 22.9%. Together, these data showed that the regulatory sequences of the *icaADBCR* locus (3′UTR of *icaR* and *icaR-icaA* IGR) were highly conserved and less prone to evolutionary changes than other IGRs.

### 3.2. Identification of Genetic Variations in the Regulatory IGRs of the icaADBCR Locus

The fact that the regulatory IGRs of the *icaADBCR* locus are highly conserved suggests that SNPs in these regions might have a significant impact on the expression of the *icaADBC* operon and on the ability of certain strains to cause PJI. To evaluate the association between particular SNPs and adaptation to PJI, we compared the sequence of *ica* regulatory IGRs of a collection of *S. aureus* clinical strains isolated from PJI (*n* = 43) and wounds (*n* = 26). We used, as a reference, the *S. aureus* MW2 genome and focused on the 3′UTR of *icaR* and the IGR between *icaR* and *icaA*. In addition, the *icaR* gene was included in the analysis since changes within the coding region might affect IcaR activity and therefore *ica* operon expression. The collection of 69 strains were clustered in seventeen groups according to the SNPs found (Figure 2). Wound isolates displayed high sequence similarity and twelve of them (46% of wound isolates) did not harbor any SNP when compared to *S. aureus* MW2 (cluster 1). On the other hand, only nine PJI isolates (21% of PJI isolates) showed a sequence identical to that of *S. aureus* MW2 (cluster 1). Cluster two included twelve PJI isolates (28% of PJI isolates) and only one wound isolate, containing a total of eleven SNPs. From these, seven SNPs were located in the 3′UTR of *icaR*, three SNPs in the 5′UTR of *icaR* and one SNP in the 5′UTR of *icaA*. Cluster three included seven PJI isolates (16% of PJI isolates) and five wound isolates (19% of wound isolates) containing five SNPs that are common to the regulatory IGRs of cluster two. Cluster four included three PJI isolates and no wound isolates, also containing a total of five SNPs that are present in IGRs of cluster two and that differ in one SNP with IGRs of cluster three. The rest of the clusters included from one to three isolates of the same origin, either PJI or wound, that contain from one to twelve SNPs, most of them present in IGRs of cluster two. Interestingly, 48 out of the total 69 strains (70% of total isolates) did not show any SNP in the promoter and 5′UTR of the *icaADBC* operon, confirming that regulation of the operon imposes strong restrictions to sequence variations. As regards the *icaR* coding sequence, fourteen synonymous and two non-synonymous SNPs were detected. Specifically, cluster five included one PJI isolate that contains one SNP that leads to a G112E change, while cluster six included two PJI isolates containing a SNP that leads to a V176E change in the carboxyl-terminal domain of the transcriptional repressor IcaR. All in all, the fact that 70% of *S. aureus* isolates from wounds clustered together with strains from PJI strongly suggests that mutations in IGRs of the *icaADBCR* locus do not contribute to adaptation to PJI.

### 3.3. Contribution of SNPs in the Regulatory IGRs of the icaADBCR Locus to PIA/PNAG Production Capacity

SNPs and small indels within IGRs can have major phenotypic consequences. To identify a biofilm phenotypic alteration associated to the SNPs found in the regulatory IGRs of the *icaADBCR* locus, we selected one PJI isolate of the most representative clusters (clusters one to four), and also one isolate of clusters five and six, encoding *icaR* mutants. Next, we determined their biofilm formation capacity in vitro, including as a control the *S. aureus* 15981 strain that forms a strong PIA/PNAG dependent biofilm [14]. All strains were very weak biofilm formers, with no significant differences in their ability to form a biofilm, except for MIC 7018 (cluster 6), that showed a very high biofilm-forming capacity (Figure 3A). Next, we analyzed the production of PIA/PNAG exopolysaccharide by dot-blot. Results showed a direct correlation between the amount of PIA/PNAG produced by each strain and its biofilm formation capacity (Figure 3B). These results suggested that the SNPs found in the regulatory IGRs of the *icaADBCR* locus do not influence the in vitro biofilm forming ability of PJI strains. Furthermore, our analyses indicated that a single mutation in the IcaR coding sequence can have a major effect on the *S. aureus* biofilm formation capacity.

To further explore the potential relationship between IGR mutations and PIA/PNAG production, we aimed at quantifying the influence of IGR mutations on transcription of the *icaADBC* genes. To this end, we constructed transcriptional fusions of the intergenic alleles amplified from each representative PJI isolate with the *gfp* gene in the pCN52 plasmid (Figure 4A). The resulting plasmids were transformed into the same recipient strain, that is the *icaADBC* mutant *S. aureus* 132 *ica::tet* [36], in order to avoid differences in genetic background that might interfere with *ica* operon expression. It is important to note that the *S. aureus* 132 strain was chosen because of its ability to produce a polysaccharidic-dependent biofilm when grown in TSB-NaCl [36]. As a positive control, we used a reporter plasmid containing the *ica* intergenic allele of the *S. aureus* 15981 strain, which includes the *icaR* gene with a synonymous mutation that results in a premature stop codon. This makes the protein less efficient, leading to a higher expression of the *ica* genes. In agreement with the above results concerning biofilm formation and PIA/PNAG production, GFP was only detectable in *S. aureus* 132 *ica::tet* carrying the reporter constructed from the cluster six isolate, which encodes for a V76E mutant in the carboxyl-terminal region of IcaR (Figure 4A). These results strongly suggested that IcaR is the dominant regulatory component in our experimental conditions and that the contribution of the SNPs in IGRs to *ica* expression, if any, might be unnoticeable in the presence of IcaR.

To assess the influence of IGRs on *ica* expression independently of IcaR, we then constructed new reporters of each intergenic allele containing a stop codon in the *icaR* coding sequence (Figure 4B). In this case, *gfp* expression was indeed detected with all the reporters. However, no relevant differences in GFP levels were observed. Together, these results demonstrated that the SNPs present in IGRs of the *icaADBCR* locus found in PJI isolates do not cause variability in the expression of the *icaADBC* operon, at least under the conditions evaluated.

### 3.4. Increasing PIA/PNAG Production Leads to a Reduction in Virulence

Next, we explored the direct consequences that de-regulation of *icaADBC* operon expression has for the capacity of *S. aureus* to colonize and survive in the host. For that, we used a *G. mellonella* infection model in which larvae were challenged with 10^7^ CFU of the *S. aureus* MW2 wild type strain and its derivative strain MW2 *icaR* V176E. The strain MW2 *icaR* V176E is an isogenic mutant of *S. aureus* MW2 containing a T to A mutation at nucleotide 527 of the chromosomal copy of *icaR*, which is the same mutation present in isolate MIC 7018 (cluster 6). This isogenic mutant strain exhibited a significantly increased PIA/PNAG-dependent biofilm formation capacity compared to the MW2 wild type strain (Figure 5). Survival assays revealed a significant reduction in virulence of MW2 *icaR* V176E strain (*p* < 0.01) compared to the wild type strain (Figure 6). These results showing that an increase in PIA/PNAG production is unfavourable for bacterial survival in the host confirmed that *ica* operon expression and PIA/PNAG production have to be tightly regulated for *S. aureus* success as a pathogen.

## 4. Discussion

PJI affects approximately 2% of arthroplasties in developed countries [44]. Most of these infections occur during the surgery procedure and originate from the microbiota of the patient’s skin. *Staphylococcus aureus* is a regular inhabitant of the human skin and is responsible for a large percentage of implant-related infections due to its capacity to colonize the surface of the implant, form a biofilm and resist the action of the immune system [45]. One question regarding the *S. aureus* strains isolated from PJI is whether they harbor adaptative mutations that make them more likely to cause the infection. In this respect, a previous study showed no differences in the clonality and prevalence of virulence genes among *S. aureus* strains isolated from PJI and nares, indicating that any clonal complex is equally prone to cause PJI [21]. Clonal complexes are established based on the nucleotide sequences of internal fragments of a standard set of metabolic housekeeping loci, and therefore, clonal complex analysis cannot detect adaptative mutations that occur within IGRs [46]. Thus, in the present work, we wondered whether *S. aureus* isolates that produce PJI contain niche adaptative mutations particularly placed at the IGRs that control the expression of the *icaADBC* operon, responsible for the production of the PIA/PNAG exopolysaccharide, a major constituent of the biofilm matrix. Interestingly, our results firstly showed that the IGR sequence between *icaR* and *icaADBC* genes is highly conserved, indicating strong restrictions on mutations of functional nucleotides [47,48,49]. The *icaR* gene and the *icaADBC* operon are transcribed in divergent orientations and therefore the IGR region encompasses the promoters, 5′ UTRs and ribosome-binding sites of both *icaR* and *icaADBC*. It has been established that within IGRs, the strength of selective constraint appears to be particularly high immediately upstream of genes [28]. However, in the case of the *icaADBCR* IGR, we observed that the SNP density decreases as a function of the distance from gene start codons, indicating that a strong selective constraint particularly affects the promoter regions of *icaR* and *icaADBC*. This is in agreement with the fact that several transcriptional regulators (SarA, TcaR, IcaR and ArlR) modulate the production of PIA/PNAG exopolysaccharide at a transcriptional level through binding to these promoter regions.

To investigate the potential functional relevance of the SNPs in the *icaADBCR* IGR to the transcription of the *icaADBC* operon, we quantified the impact of a subset of intergenic mutations on the transcription of the *gfp* reporter gene, using transcriptional fusions of both a control sequence and mutant intergenic alleles. The results showed that mutations in the *icaADBCR* IGR do not affect the transcriptional levels of the downstream *icaADBC* operon. Therefore, the polymorphisms found do not appear to be involved in the regulatory evolution of PIA/PNAG synthesis that might favour biofilm development. On the contrary, our study revealed that a single mutation in the coding region of *icaR* causes a significant increase in PIA/PNAG production. Jeng et al. previously described significant differences in the IcaR-binding affinity to its cognate DNA domain when some N-terminal residues (Leu23, Lys33 and Ala35) were substituted with those present in other TetR family transcriptional repressors [50]. To the best of our knowledge, this is the first description of a single aminoacid substitution in the carboxyl-terminal domain of IcaR that leads to a significant increase in PIA/PNAG production. Since the carboxyl-terminal domain of IcaR is involved in protein dimerization [50], further studies should be conducted to clarify the structural reasons behind the PIA/PNAG hyper-producer phenotype linked to the V176E mutation.

To evaluate the consequences of harbouring such an IcaR allele, we mutated the *icaR* gene in the *S. aureus* MW2 reference strain and performed a virulence assay using the waxworm *G. mellonella* model. With this approach, we were able to demonstrate that the single *icaR* V176E mutation reduces in vivo fitness and, therefore, increases the survival of *G. mellonella*. These results are in agreement with previous studies showing that PIA/PNAG overproduction is associated with a high fitness cost [51]. In particular, Brooks & Jefferson found that an isogenic PIA/PNAG-negative MN8 strain shows an in vitro fitness gain compared to the PIA/PNAG-overproducing MN8 strain. Similarly, non-mucoid mutants were selected over time from cystic fibrosis patients firstly colonized with highly mucoid *S. aureus* clones that over produced PIA/PNAG [52].

Altogether, these results support the view that mutations inside coding genes may increase the fitness in a particular trait (in this case, biofilm formation) at the expense of a fitness decrease in another trait (in this case, virulence). On the other hand, adaptative mutations in noncoding DNA may favour subtle changes in the expression of downstream genes while maintaining responsiveness to environmental cues, such as those present on the implanted prosthesis [53].

We acknowledge some limitations of our study such as the further need to evaluate the role of adaptative mutations upstream of genes encoding for transcriptional regulators of the *icaADBC* operon. In addition, in the future, this analysis should be extended to the IGRs flanking all genes that encode for proteins involved in biofilm development and the colonization of prostheses.

## 5. Conclusions

In conclusion, regulatory sequences of the *icaADBCR* locus are highly conserved among clinical isolates from PJI and wounds. SNPs in IGRs of the *ica* operon did not have any impact in *ica* operon expression in the strains under study and in the evaluated conditions. Furthermore, our results demonstrated that an increase in PIA/PNAG production is detrimental for *S. aureus* virulence, and that variations in the capacity to produce a biofilm can be due to single aminoacid changes in the IcaR protein.

## Figures and Tables

**Figure 1 microorganisms-10-00600-f001:**
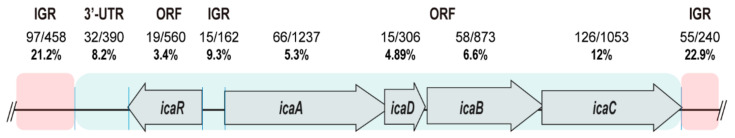
Analysis of nucleotide variation rates along the *icaADBCR* locus. The 3′UTR of *icaR* and the *icaR*-*icaA* IGR show nucleotide variation rates similar to *ica* coding sequences. The flanking IGRs, highlighted in red boxes, show high variation rates. The nucleotide variation rate in each region was calculated using 1000 genomic sequences available at the NCBI web page and the percentage represents the ratio between the total number of nucleotide changes in at least one *S. aureus* genome, and the length of the analysed sequence. ORF, open reading frame; IGR, intergenic region.

**Figure 2 microorganisms-10-00600-f002:**
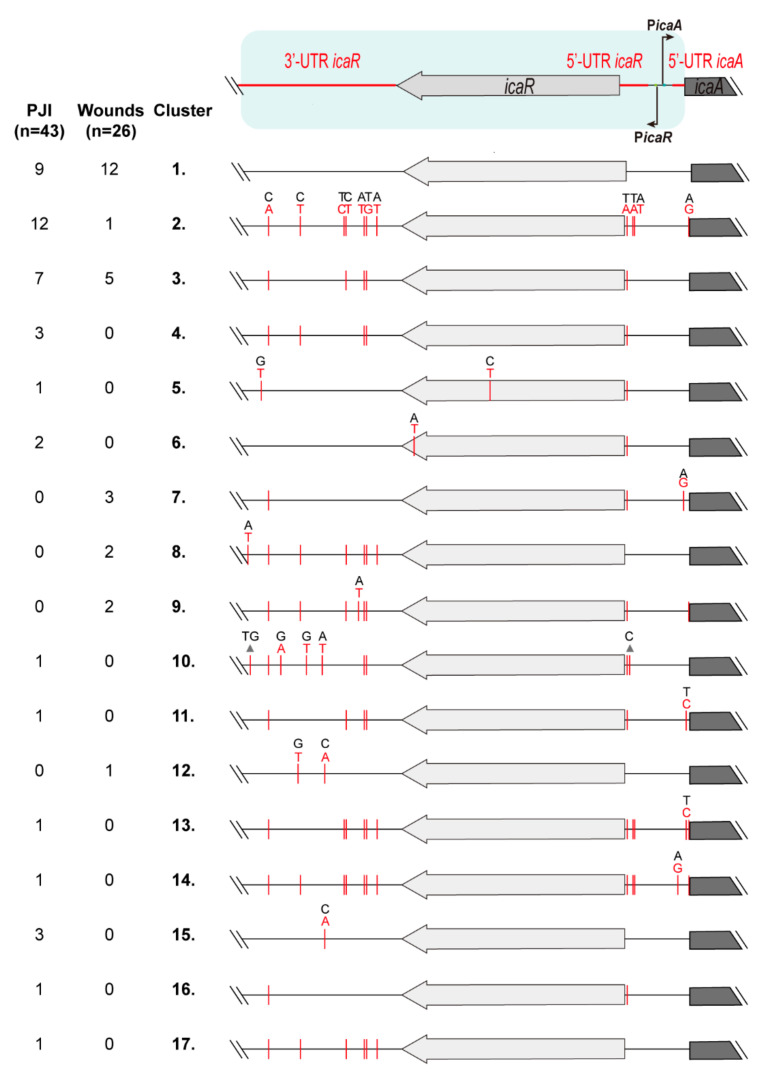
Analysis of genetic variations in IGRs controlling expression of the *icaADBCR* locus. PJI and wound isolates were grouped into seventeen different clusters according to SNPs present in the 3′UTR of *icaR*, the *icaR* coding sequence, and the IGR between *icaR* and *icaA* when compared to the sequence of the reference strain MW2 (cluster 1). The red lines show the SNPs or indels found. All sequence variations found in cluster two are depicted (nucleotide changes from a black to a red nucleotide in the upper strand). In the rest of the clusters, only the sequence variations that are different from the ones found in cluster two are detailed.

**Figure 3 microorganisms-10-00600-f003:**
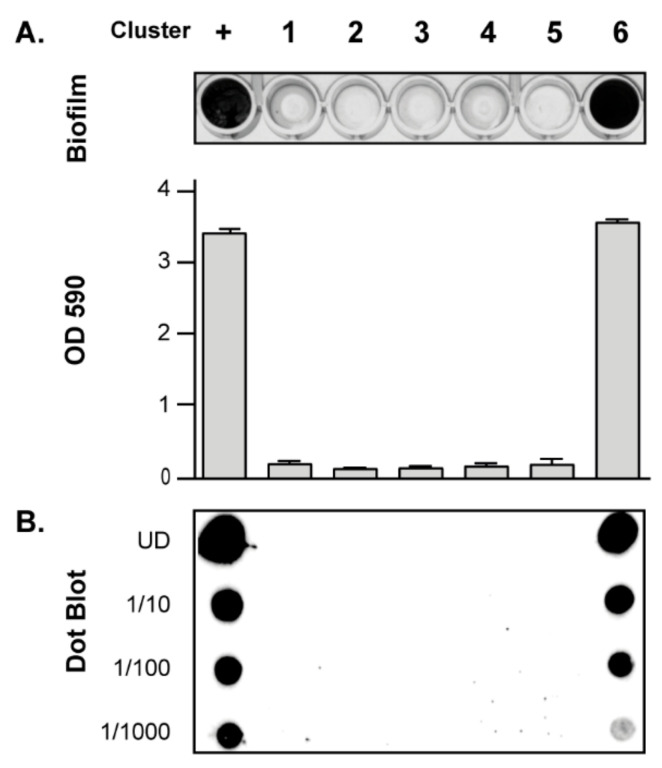
Biofilm formation capacity and PIA/PNAG production analyses of one representative *S. aureus* PJI isolate of clusters one to six. Cluster 1: MIC 6935; cluster 2: MIC 6924; cluster 3: MIC 6934; cluster 4: MIC 6936; cluster 5: MIC 6948; cluster 6: MIC 7018. (**A**) Biofilm phenotype on polystyrene microtiter plates after 24 h of growth at 37 °C in TSB-NaCl medium. Bacterial cells were stained with crystal violet, and biofilms were quantified by solubilizing the crystal violet with alcohol-acetone and determining the absorbance at 590_nm_. The error bars represent the standard deviations of the results of three independent experiments. (**B**) Quantification of PIA/PNAG exopolysaccharide biosynthesis by dot-blot. Samples were analyzed after 16 h of static incubation in TSB-NaCl at 37 °C. Serial dilutions (1/10) of the samples were spotted onto nitrocellulose membranes and PIA/PNAG production was detected with specific anti-PIA/PNAG antibodies. UD; undiluted sample. +; a strong PIA/PNAG dependent biofilm forming strain was included as a control.

**Figure 4 microorganisms-10-00600-f004:**
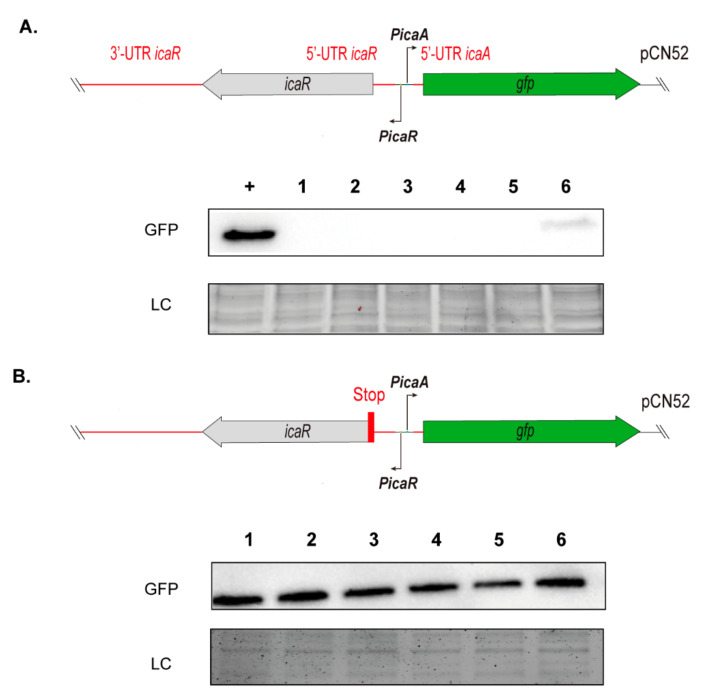
GFP protein levels in *S. aureus* 132 *ica::tet* expressing *ica* intergenic alleles-*gfp* reporter fusions. (**A**) Schematic representation of the transcriptional fusions of the *ica* intergenic alleles amplified from each representative PJI isolate with the *gfp* gene in the pCN52 plasmid. The Western blot shows GFP protein levels in the *S. aureus* 132 *ica::tet* strain expressing the reporter fusions. +; a reporter plasmid containing the *ica* intergenic allele of the *S. aureus* 15981 strain was used as a positive control. LC; a stain-free gel portion is included as loading control. (**B**) Analysis using reporter fusions containing a stop codon in the *icaR* coding sequence.

**Figure 5 microorganisms-10-00600-f005:**
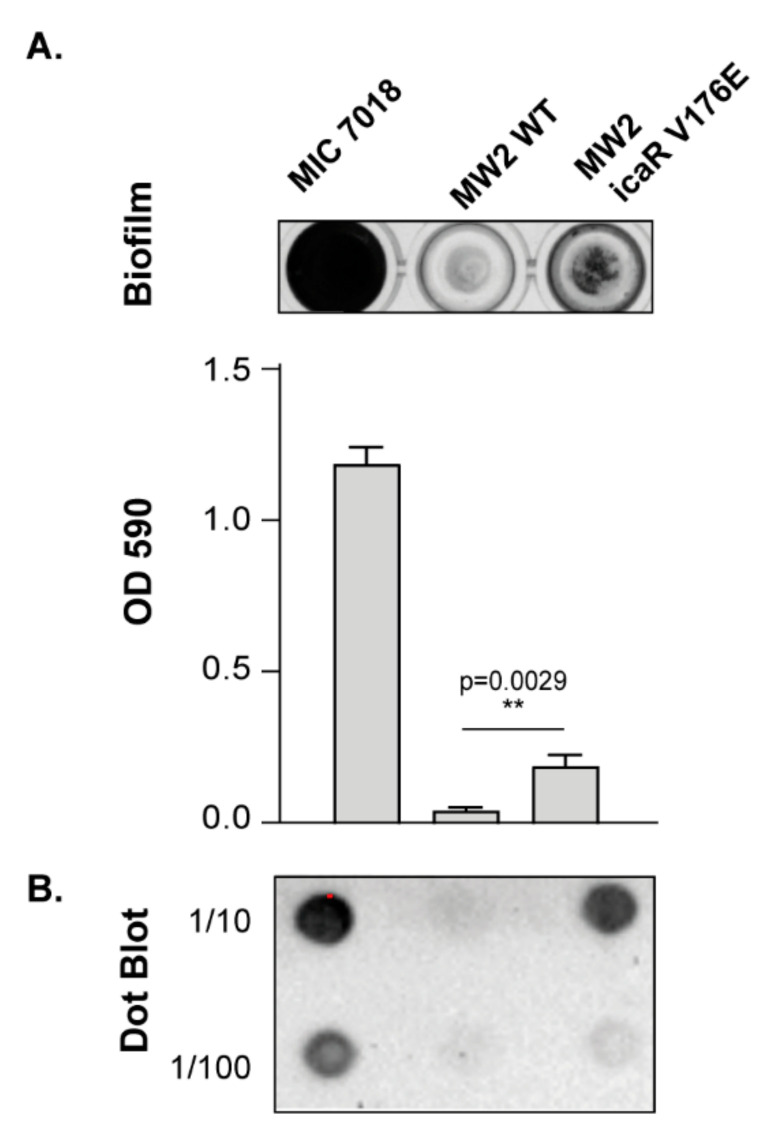
Biofilm formation capacity and PIA/PNAG production of the MW2 *icaR* V176E mutant. (**A**) For comparison, the MIC 7018 strain (cluster 6) and the MW2 wild type strains were included in the analysis. Biofilm phenotype on polystyrene microtiter plates was visualized after 24 h of growth at 37 °C in TSB-NaCl medium. The biofilm was stained with crystal violet, and the biofilm biomass was quantified by solubilizing the crystal violet with alcohol-acetone and determining the absorbance at 590_nm_. The data are shown as mean ± SD of three independent experiments. Statistical analysis was performed by two-tailed unpaired *t*-test. (**B**) Quantification of PIA/PNAG exopolysaccharide biosynthesis by dot-blot. Samples were analysed after 16 h of static incubation in TSB-NaCl at 37 °C. Serial dilutions (1/10) of the samples were spotted onto nitrocellulose membranes and PIA-PNAG production was detected with specific anti-PIA-PNAG antibodies.

**Figure 6 microorganisms-10-00600-f006:**
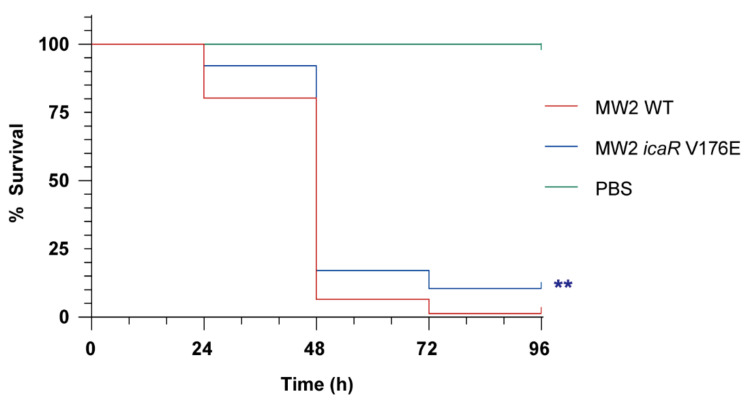
*Galleria mellonella* infection with *S. aureus* MW2 wild type and MW2 *icaR* V176E mutant strains. Groups of larvae (25/group) were inoculated with 10 μL PBS (uninfected control group) or bacterial suspensions containing 10^7^ CFU of the corresponding strain, into the last proleg and incubated at 37 °C. Worms were checked daily, and any deaths were recorded every 24 h, for a total of four days. Three independent experimental trials were performed. Survival data were plotted using the Kaplan-Meier method and expressed as a percentage of survival versus time. Statistically significant differences were determined using the log rank test (**, *p* < 0.01).

**Table 1 microorganisms-10-00600-t001:** Bacterial strains used in this study.

Strain	Relevant Characteristic(s)	MIC ^a^	Reference
*E. coli* IM01B	*E. coli* K12 DH10B Δ*dcm* and containing the *hsdS* gene of MW2 integrated between the *essQ* and *cspB* genes. Used for cloning experiments and isolation of plasmids that are transformed into *S. aureus* CC1 strains at high efficiency	5694	[34]
*S. aureus* 15981	MSSA clinical isolate; strong biofilm producer; PNAG-dependent biofilm matrix	0532	[14]
*S. aureus* MW2	Community-acquired strain of MRSA, which was isolated in 1998 in North Dakota, USA	3566	[35]
*S. aureus* MW2 *icaR* V176E	MW2 strain harbouring a mutation in the *icaR* gene, coding for a V176E IcaR variant	7983	This study
*S. aureus* 132 *ica::tet*	*S. aureus* 132 containing a tetracycline resistance cassette that replaces *ica* genes	3343	[36]
*S. aureus* clinical isolate	Sahlgrenska University Hospital (Sweden). Periprosthetic joint infection (hip)	6924	[32,33]
*S. aureus* clinical isolate	Sahlgrenska University Hospital (Sweden). Periprosthetic joint infection (hip)	6934	[32,33]
*S. aureus* clinical isolate	Sahlgrenska University Hospital (Sweden). Periprosthetic joint infection (hip)	6935	[32,33]
*S. aureus* clinical isolate	Sahlgrenska University Hospital (Sweden). Periprosthetic joint infection (hip)	6936	[32,33]
*S. aureus* clinical isolate	Sahlgrenska University Hospital (Sweden). Periprosthetic joint infection (hip)	6948	[32,33]
*S. aureus* clinical isolate	Sahlgrenska University Hospital (Sweden). Periprosthetic joint infection (knee). Strong biofilm producer	7018	[32,33]
*S. aureus* clinical isolates	Culture Collection at University of Gothenburg, (CCGU), Sweden. Wounds	7032–7050 7166–7172	This study

^a^: Identification number of each strain in the culture collection of the Laboratory of Microbial Pathogenesis, Navarrabiomed-*Universidad Pública de Navarra*.

**Table 2 microorganisms-10-00600-t002:** Plasmids and oligonucleotides used in this study.

Plasmids	Relevant Characteristics	Reference
pJET1.2	Cloning vector. Amp^R^.	Thermo Scientific
pMAD	*E. coli-S. aureus* shuttle vector with a thermosensitive origin of replication for Gram-positive bacteria. Amp^R^ Ery^R^	[37]
pMAD::*icaR*V176E	pMAD plasmid containing the DNA sequence for *icaR* V176E mutation	This study
pCN52	*E. coli*-*S. aureus* shuttle vector with promoterless *gfpmut2*. Ery^R^	[38]
pCN52::IGR*_ica_* C1	3′UTR *icaR-icaR—*IGR *icaR*/*icaA* amplified from MIC 6935 and fused to promoterless *gfpmut2* in pCN52	This study
pCN52::IGR*_ica_* C2	3′UTR *icaR-icaR—*IGR *icaR*/*icaA* amplified from MIC 6924 and fused to promoterless *gfpmut2* in pCN52	This study
pCN52::IGR*_ica_* C3	3′UTR *icaR-icaR—*IGR *icaR*/*icaA* amplified from MIC 6934 and fused to promoterless *gfpmut2* in pCN52	This study
pCN52::IGR*_ica_* C4	3′UTR *icaR-icaR—*IGR *icaR*/*icaA* amplified from MIC 6936 and fused to promoterless *gfpmut2* in pCN52	This study
pCN52::IGR*_ica_* C5	3′UTR *icaR-icaR—*IGR *icaR*/*icaA* amplified from MIC 6948 and fused to promoterless *gfpmut2* in pCN52	This study
pCN52::IGR*_ica_* C6	3′UTR *icaR-icaR—*IGR *icaR*/*icaA* amplified from MIC 7018 and fused to promoterless *gfpmut2* in pCN52	This study
pCN52::IGRStop*_ica_* C1	3′UTR *icaR-icaR_STOP_—*IGR *icaR*/*icaA* amplified from MIC 6935 and fused to promoterless *gfpmut2* in pCN52	This study
pCN52::IGRStop*_ica_* C2	3′UTR *icaR-icaR_STOP_—*IGR *icaR*/*icaA* amplified from MIC 6924 and fused to promoterless *gfpmut2* in pCN52	This study
pCN52::IGRStop*_ica_* C3	3′UTR *icaR-icaR_STOP_—*IGR *icaR*/*icaA* amplified from MIC 6934 and fused to promoterless *gfpmut2* in pCN52	This study
pCN52::IGRStop*_ica_* C4	3′UTR *icaR-icaR_STOP_—*IGR *icaR*/*icaA* amplified from MIC 6936 and fused to promoterless *gfpmut2* in pCN52	This study
pCN52::IGRStop*_ica_* C5	3′UTR *icaR-icaR_STOP_—*IGR *icaR*/*icaA* amplified from MIC 6948 and fused to promoterless *gfpmut2* in pCN52	This study
pCN52::IGRStop*_ica_* C6	3′UTR *icaR-icaR_STOP_—*IGR *icaR*/*icaA* amplified from MIC 7018 and fused to promoterless *gfpmut2* in pCN52	This study
**Oligonucleotides**	**Sequence** ** ^a^ **	
Au56	ATGCCTGCAG**GTCGA**CCGAGTAGAAGCATCATCATTACTTGATT	
Au73	**GGATCC**TAAGCCATATGGTAATTGATAG	
Au76	ACGAATTCGAGCTC**GGTA**CCTTTCTTTACCTACCTTTCGTTAGTTAGGTTG	
Au78	TTATTGATAACGCAATAACCTTATAAGGATCCTTTTCAGAGAAGGGGTATGACGG	
Au77	CCGTCATACCCCTTCTCTGAAAAGGATCCTTATAAGGTTATTGCGTTATCAATAA	
LM9	CAAAGATGAAG**AGTACT**CGCTACTAAATA	
LM10	TAGTAGCG**AGTACT**CTTCATCTTTGAATTG	

^a^: Restriction enzymes sites and nucleotides for gene editing are indicated in bold and underlined, respectively.

## Data Availability

All datasets generated for this study are included in the article. Genome sequence with detailed information of the strains is available in the National Center for Biotechnology Information (NCBI) under the BioProject accession number PRJNA765573.

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
