# Peer review of "Experimental Polymorphism Survey in Intergenic Regions of the icaADBCR Locus in Staphylococcus aureus Isolates from Periprosthetic Joint Infections"

_microorganisms, 2022, doi:10.3390/microorganisms10030600_

Round 1

Reviewer 1 Report

The manuscript submitted for evaluation is a very interesting description of research into the molecular aspects of biofilm formation. The manuscript is written in accordance with applicable rules. Particularly interesting is the part concerning the obtained research results. Discussion and summary correlate with the obtained results. The work may be printed without corrections. 

Reviewer 2 Report

General comment: The authors have covered and described an interesting topic and it is a beautiful paper! But in some places, it is not clear due to the chosen English. Please find critical comments as follows:

Critical comments and suggestions:

Title: The referee suggests to improve the title, it is long and not clear.

Abstract: This section isn’t very clear; it needs also to be improved in English.

1. Introduction: This section is very confusing and the referee suggests reviewing it following a chronological path.

- Aim: The aim must be a new paragraph starting on line 80 and to be clear.  

- Lines 82-91: These sentences seem like results and conclusions; why here?

2. Materials and Methods

- Lines 102: “LB and TSB”, need to put also city and state of the manufacturer where the materials were taken from.

- Line 116: You need to express as follow this name “STAB VIDA, Lda.”

- Line 118: Put the city and state of the manufacturer Thermo Scientific.

- Line 120: Use the acronym of E. coli here, because it is used for the first time as a long name on line 101.

- Line 142: City and state of Illumina.

- Line 171: The referee suggests to use the acronym of OD and replacing it in the following text.

- Line 172: Also, here the city and the state of the material used. This must be added for all materials used in the entire manuscript.

- Line 213: After the number needs a space, before the unit. Use “L”  for 0.3 mL, because “L” is used during the entire manuscript.

- Line 218: The referee suggests to delete this phrase, because is repeated in the subsection 2.8.

- The referee suggests reviewing the Results and Discussion according to the English used here, because the sentences are not very clear.

Round 2

Reviewer 2 Report

The authors modified and improved the manuscript based on the comments made. But the referee has one final suggestion for the authors to change before publication. On line 96, please express in this way "in order to explore". On line 98, the authors can start the sentence with "We intend to show" and deleting "Our results...", because we are in the aim yet. In line 99, please delete "suggesting that the necessity" and continue the sentence with "necessary"...correcting the rest of the sentence. On line 101, please put "aimed to evidence" instead of "provide evidence"...and then correct the rest of the sentence.

Author Response

Reviewer 2

The authors modified and improved the manuscript based on the comments made. 

Response: We would like to thank the reviewer for his/her helpful comments to improve the manuscript.

But the referee has one final suggestion for the authors to change before 
publication. On line 96, please express in this way "in order to explore". On 
line 98, the authors can start the sentence with "We intend to show" and 
deleting "Our results...", because we are in the aim yet. In line 99, please 
delete "suggesting that the necessity" and continue the sentence with 
"necessary"...correcting the rest of the sentence. On line 101, please put 
"aimed to evidence" instead of "provide evidence"...and then correct the rest 
of the sentence.

Response: We respectfully disagree with the reviewer’s suggestions. We have used a common writing style that includes a few sentences at the end of the introduction to give a preview of the main results and to state the contribution of our work. We believe that including the changes suggested by the reviewer modifies the meaning of the sentences (see below). We therefore ask to maintain the original paragraph.

Original paragraph

Here, we aimed at identifying SNPs in the regulatory IGRs of the icaADBCR locus in the genomes of a collection of S. aureus isolates from PJI, and at analysing their potential association with the propensity to cause arthroplasty infection. Using reporter fusions of representative IGRs with the green fluorescent protein encoding gene, we explored the contribution of SNPs in IGRs to icaADBC operon transcription. Also, PIA/PNAG production and biofilm formation capacity of representative isolates was examined. Our results show high sequence conservation of IGRs, suggesting that the necessity to tightly regulate PIA/PNAG expression imposes constraints to IGRs of the icaADBCR locus. We further provide evidence that the SNPs found in PJI isolates are not associated with changes in icaADBC expression and the capacity of the strains to synthesize PIA/PNAG, form a biofilm and cause PJI.  

Paragraph including reviewer’s 2nd suggestion

Here, we aimed at identifying SNPs in the regulatory IGRs of the icaADBCR locus in the genomes of a collection of S. aureus isolates from PJI, and at analysing their potential association with the propensity to cause arthroplasty infection. Using reporter fusions of representative IGRs with the green fluorescent protein encoding gene, in order to explore the contribution of SNPs in IGRs to icaADBC operon transcription. Also, PIA/PNAG production and biofilm formation capacity of representative isolates was examined. We intend to show high sequence conservation of IGRs, necessary to tightly regulate PIA/PNAG expression imposes constraints to IGRs of the icaADBCR locus. We further aimed to evidence that the SNPs found in PJI isolates are not associated with changes in icaADBC expression and the capacity of the strains to synthesize PIA/PNAG, form a biofilm and cause PJI.